# Autochthonous Apple Cultivars from the Campania Region (Southern Italy): Bio-Agronomic and Qualitative Traits

**DOI:** 10.3390/plants12051160

**Published:** 2023-03-03

**Authors:** Danilo Cice, Elvira Ferrara, Anna Magri, Giuseppina Adiletta, Giuseppe Capriolo, Pietro Rega, Marisa Di Matteo, Milena Petriccione

**Affiliations:** 1CREA, Council for Agricultural Research and Economics, Research Centre for Olive, Fruits and Citrus Crops, Via Torrino 3, 81100 Caserta, Italy; 2Department of Environmental Biological and Pharmaceutical Sciences and Technologies, University of Campania “Luigi Vanvitelli”, Via Vivaldi 43, 81100 Caserta, Italy; 3Department of Industrial Engineering, University of Salerno, Via Giovanni Paolo II, 84084 Fisciano, Italy

**Keywords:** fruit crop, germplasm, ancient cultivars, statistical analysis

## Abstract

Apple (*Malus* × *domestica* Borkh.) is an important fruit crop widely spread in the cold and mild climates of temperate regions in the world, with more than 93 million tons harvested worldwide in 2021. The object of this work was to analyze thirty-one local apple cultivars of the Campania region (Southern Italy) using agronomic, morphological (UPOV descriptors) and physicochemical (solid soluble content, texture, pH and titratable acidity, skin color, Young’s modulus and browning index) traits. UPOV descriptors highlighted similarities and differences among apple cultivars with a depth phenotypic characterization. Apple cultivars showed significant differences in fruit weight (31.3–236.02 g) and physicochemical trait ranging from 8.0 to 14.64° Brix for solid soluble content, 2.34–10.38 g malic acid L^−1^ for titratable acidity, and 15–40% for browning index. Furthermore, different percentages in apple shape and skin color have been detected. Similarities among the cultivars based on their bio-agronomic and qualitative traits have been evaluated by cluster analyses and principal component analyses. This apple germplasm collection represents an irreplaceable genetic resource with considerable morphological and pomological variabilities among several cultivars. Nowadays, some local cultivars, widespread only in restricted geographical areas, could be reintroduced in cultivation contribution to improving the diversity of our diets and contemporary to preserve knowledge on traditional agricultural systems.

## 1. Introduction

Apple (*Malus* × *domestica* Borkh.) is the major fruit crop worldwide that grows in temperate regions with both cold and moderate climates [1]. The total world apple production amounted to 93 million tons, with a production area of 4.8 million hectares in 2021, being one of the economically most important fruit crops. Italy is the eighth largest producer in the world and second in Europe, accounting for the production of about 2.2 million tons in 2021 [2]. As reported by the Second Report on the State of the World’s Plant Genetic Resources for Food and Agriculture apple represents the second fruit crop with the largest number of accessions and several ex-situ collections in the world [3]. Italy adheres to the Food and Agriculture Organization (FAO) international undertaking on plant genetic resources, which has as its primary goals the discovery, collection, conservation and identification of endangered plant genetic resources for food and agriculture [4]. Germplasm collections represent one of the three principal ways of germplasm management, and it is the most effective long-term investment for ex-situ conservation to preserve the genetic diversity and agronomic characteristics of the endangered local cultivars of different fruit crops [5].

The preservation and the study of germplasm collections assume an important role in breeding programs to evaluate new traits that can be introgressed into new cultivars [6,7]. Over the years, the development of intensive cultivation of fruit crops and the preference for fruits with uniform shapes and sizes that meet the market standards have promoted the gradual replacement by farmers of traditional varieties with new, improved cultivars [8]. The widespread use of these cultivars has caused in uniformity of commercial apple orchards and the shrinking of genetic diversity in the global markets; consequently, particular attention was paid to the analysis of apple collections maintained in national and local repositories worldwide [9,10,11,12]. Nowadays, autochthonous cultivars are regionally cultivated mostly in marginal areas such as solitary trees, field boundaries or in small orchards, and represent the local germplasm [13]. They exhibit good local environmental conditions and are genetic resources with high genetic variability that can detect resistance to biotic and abiotic stress, as well as interesting phenological and quality traits. Traditional apple cultivars with particular flavors and tastes are not grown for mass production but for local market consumption or for local peculiarity in use or processing after harvest [4,14,15]. The autochthonous genetic heritage of fruit species can be represented by low-input cultivars with relatively stable production even under extreme conditions, and the interest in their depth characterization is growing [4].

Morphological and bio-agronomic characterization of Malus has been applied for accurate descriptions of germplasm collections, breeding programs [16] and taxonomic studies [17,18], although these descriptors are strongly affected by the age of trees, the training systems, and the plants’ environmental conditions [19]. The most utilized morphological descriptors are derived from international guidelines such as the International Board for Plant Genetic Resources (IBPGR) [20] and the Union for the Protection of New Varieties of Plants (UPOV) [21]. Several studies have demonstrated that local apple cultivars collected in Bosnia and Herzegovina [22], Canada [23], Hungary [24], India [25], Iran [26], Italy [7,27], Kashmir valley [28], Macedonia [29], Montenegro [16], Serbia [30] and Turkey [31], showed high variability in morphological traits.

Italy is included as a category 2 country with medium to high agro-biodiversity [4], which belongs to the classical Mediterranean center of diversity already recognized by Vavilov [32]. Two different studies conducted on plant genetic resources exploration in the same area of Southern Italy have demonstrated that about 75% of the landraces have been lost from 1950 to 1980 [33,34]. Apple production has a long history in the Campania region due to its significance for the local economy and history, and high levels of phenotypic diversity among local apple cultivars have also been found [7,8,35].

In this study, 31 ancient apple cultivars at risk of extinction collected in the different provinces of the Campania region (Southern Italy) were characterized based on morphological and bio-agronomic traits. Furthermore, qualitative parameters such as total soluble solids content (°Brix), juice pH, titratable acidity, firmness and Young’s module were performed.

## 2. Results and Discussion

### 2.1. Morphological, Pomological and Physicochemical Traits

The geographical position of the original apple plants from which budwood was collected to realize the ex-situ germplasm collection, located in Pignataro Maggiore (Southern Italy), owned by the CREA-OFA, are shown in Figure 1.

Traditional endangered varieties have been detected in several areas also marginal in the Campania region known as “Campania Felix” until Roman civilization times due to its agricultural tradition. Different environmental and phytoclimatic conditions have contributed to the formation of particular fruit culture contexts with specialized and traditional cultivation with a high degree of biodiversity [36].

Autochthonous apple cultivars were described using 57 morphological traits descriptor as established by Union for the Protection of New Varieties of Plants (UPOV) guidelines (TG/14/9-UPOV 2005) [21] and compared with two standard cultivars ‘Annurca Rossa del Sud’ and ‘Golden B’ cultivated in the same field (Table 1, Table 2, Appendix A and Appendix A).

The first group of bio-agronomic traits, which was employed to assess the apple cultivars, involved nine tree characters. Tree vigor is medium in the majority of apple cultivars (69.69%), followed by strong vigor (30.30%; Appendix A). Tree vigor plays an important role in orchard management as it can affect the within-canopy microclimate and, consequently, disease development or fruiting patterns [37].

Ramified tree types with upright and spreading habits were the predominant (93.93%) in most of the apple cultivars with three different types of bearing: on spur only (18.18%), on spurs and long shoots (39.39%) and on long shoots only (42.42%). One-year-old shoots show thin or medium thickness (90.90%), and only in ‘Parrocchiana’ are very thick, while the medium length of the internode in 69.70% of analyzed apple cultivars with reddish brown color (72.72%) on the sunny side. On the distal half of shoots was observed a medium or strong pubescence (75.76%) with small numbers of lenticels (60.61%; Appendix A).

The second group of descriptors used to evaluate the apple cultivar included eight, six, and 34 traits of leaves, flowers and fruits, respectively. Among the analyzed apple cultivars, about 91% possessed an upwards leaf attitude in relation to the shoot. The leaf blade shows medium length in 33.33% of apple cultivars, followed by a short length in 30.30% of selected trees, while the leaf length was mostly narrow (42.42%). A medium-intensity green color (90.91%) was observed in the leaf blade with incisions on the margin of serrate type 2 (39.39%) and medium pubescence on the lower side (51.51%). The petiole was long in 36.36% of the analyzed apple cultivars, with a large extent of anthocyanin coloration from the base (69.70%). The predominant color at the balloon stage in the flowers is light pink (54.54%) with a medium diameter (96.97%) and free (45.45%) or intermediate (42.42%) arrangement of petals. Many apple cultivars have flowers with the elongation of the style above the anthers (51.51%) with a medium time of the beginning of flowering in 45.45% of the analyzed cultivars. Young fruits show an absent or small extent of red overcolor due to low anthocyanin content (87.88%; Appendix A). The tree size and vegetative growth are influenced by environmental as well as genetic factors in several fruit crops [38].

The fruits’ general fruits shape is obloid (42.42%), followed by conic (30.30%) and globose (18.18%) shapes with absent or weak ribbing (93.94%), crowning at the calyx end (75.75%) and greasiness of skin (69.70%) in the most of apple cultivars (Appendix A). Centuries of domestication, selection, and propagation have resulted in a wide range of fruit shapes among apple cultivars [13]. Arnal et al. [39] demonstrated that fruit shape was mostly conical, with a minority of ellipsoidal and flat globose shapes in apple germplasm collected in Central Spain.

In this study, the majority of cultivars (75.75%) have a yellow-green skin ground color, and 54% of these showed skin with medium intensity and different patterns that varied from pink-red to a red color, as demonstrated in other studies carried out to Arnal et al. [39], Bozovic et al. [16], and Zovko et al. [40] (Appendix A). Skin color in apples is due to several secondary metabolites; in particular, anthocyanins concentration is regulated by genetic and environmental factors, and it has been identified as a transcription factor (MYB1) as a key regulator for red pigmentation in the skin [41]. Instead, the two main environmental factors that affect apple red skin coloration development are temperature and light [42].

In most of the apple fruits, we did not observe an area of russet around stalk attachment (66.67%), on cheeks (84.84%) and around the eye basin (81.81%). Lenticels on apple skin are few, with small size in 54.54% of the selected fruits, while the stalk has a medium length (60.60%) and thickness (69.69%) with narrow width and medium depth of both the stalk cavity and the eye basin in the most of analyzed fruits. White flesh is displayed in 84.84% of apple cultivars, while the others have cream-colored flesh (Appendix A). Most of these parameters as well as size, color, the presence or lack of imperfections, and/or the presence of russet, affect customer choice, making them key breeding goals [43].

According to ANOVA (*p* ≤ 0.05), most traits measured showed significant differences among the investigated accessions. Young fruit: the extent of anthocyanin overcolor showed the greatest CV (103.89%) and followed by the relative area of overcolor (84.07%), size (82.44%), the pattern of overcolor (77.84%), ratio height/diameter (69.22%) and area of russet around eye basin (52.49%), respectively. In addition, 52 out of 57 characters (91.23% in total) showed CVs greater than 20.00%, indicating a high variation among the accessions. In contrast, five out of 57 traits measured showed CVs less than 20.00% (Table 1).

Bio-agronomic and qualitative traits of all apple genotypes are shown in Table 2, while ANOVA results are reported in Appendix A. The highly significant F-values for fruit weight and physicochemical features (*p* < 0.001) suggested that apple genotypes accounted for a significant proportion of variance in the dependent traits. A strong effect for genotype in explaining the variance (≥90.0%) in dependent variables was confirmed by Omega squared (ω^2^) values (Appendix A). Furthermore, the Duncan test allowed us to distinguish among genotypes for most traits.

In our study, several cultivars have a wide harvesting season; fruit production occurred throughout the 5-month period of evaluation, with very early-ripening apple cultivars in late June, such as ‘Acquata’, ‘Aitaniello’, and ‘San Giovanni’ and very late cultivars in late October such as ‘Ambrosio’ and ‘Re’. The distribution frequencies of harvest date were 12.1%, 3.0%, 24.2%, 51.5% and 9.1% in very-early, early, medium, late, and very late ripening, respectively (Appendix A).

Significant differences in the fruit weight, skin color, flesh firmness, SSC, TA and pH were observed among the analyzed cultivars. The fruit weight changed significantly among the apple genotypes, with a minimum of 31.33 ± 5.71 g in ‘Carne’ and a maximum of 236.02 ± 31.63 g in ‘Tubiona’. Most of the cultivars (51.6%) showed an average fruit weight ranging from 31.33 to 79.20 g (Table 2; Appendix A). In a previous study carried out on ancient apple cultivars in the region of Garfagnana (Tuscany, Italy), has been demonstrated that fruit weight ranged from 88.8 g in ‘Lugliese Grisanti’ to 256.0 g in ‘San Michele’. Furthermore, in other studies, fruit weight showed a maximum value of 81.34 g in Sel-20 and 310.99 in Vanel-071 and a minimum of 1.06 g in Sel-3 and 43.04 in Vanel-137 in apple collection in the Kashmir Region and in Turkey [44,45].

SSC is an important trait useful to measure the level of dissolved sugars in apples due to the hydrolysis of starch and to check fruit ripeness, which is important to determine harvest timing [46]. Furthermore, the levels of sugars and organic acids determine the taste of ripe fleshy fruit, and the interaction of sugar and acid metabolism influences the relative content of these constituents [47]. In apples, the most abundant soluble sugars are fructose and sucrose, while malic acid is the main organic acid that accounts for up to 90% of the total organic acids [48]. ‘Chianella’ showed the highest SSC (14.64 ± 0.30 Brix) while ‘San Giovanni’ was the lowest one (8.00 ± 0.30 Brix; Table 2). Similar studies reported that SSC varied from 9.0 to 18.1 for apple genotypes or selections from several countries [44,45,49,50]. TA displayed a higher value (10.38 ± 0.63 g malic acid L^−1^) in ‘Vivo’ while the lowest statistically significant values were found in ‘Paradiso’ (2.34 ± 0.49 g malic acid L^−1^) and ‘Parrocchiana’ (2.34 ± 0.12 g malic acid L^−1^). As reported by Zhang and Han [48], the malic acid content ranged from 0.5 to 18.9 mg/g in cultivated apples. Depending on the type of cultivar, the pH levels varied greatly. The lowest pH (2.67 ± 0.15) was observed for ‘San Francesco’, whereas the highest one (5.28 ± 0.15) was observed for ‘Paradiso’. Several studies have demonstrated that pH values ranged from 3.15 to 4.89 for apple genotypes [44,51].

Firmness is a quality indicator of fresh and processed agricultural products evaluated by all investors in the production chain [52]. The firmness values are used to predict the manipulation resistance of fruits [14]. In apples, it is generally established that the lowest value of accepted fruit firmness is 6–7 kg cm^−2^. Lower values were recorded for the ‘Cannamela’, ‘Carne’, ‘Cusanara’, ‘Prete’, and ‘San Nicola’, while higher values were registered for the other cultivars showing firmness ranging from 8 to 10 kg cm^−2^ (Table 2). The results showed that some traditional cultivars have resistances that are statistically similar, if not higher, than standard cultivars. This means that most traditional apple cultivars are resistant to mechanical stresses or normal physiological processes, as reported by Panzella et al. [14] for other local Italian apple cultivars. Great variability in flesh firmness has been highlighted among genotypes collected from apple germplasm in Turkey and Kashmir region [44,45]. Firmness variations are due only to genetic variability among the analyzed cultivars cultivated in the same pedo-climatic conditions [48].

Young’s modulus is a key indicator to describe flesh texture in fruit crops (Sinha and Bhargav, 2020), ranging from 0.3 to 2.15 MPa in analyzed apple cultivars. The highest Young’s Modulus was detected in ‘Tenerella’ and ‘Re’ without significant differences with ‘Annurca Rossa del Sud’ while the lowest ones were measured in ‘Acquata’, ‘Aitaniello’, and ‘San Giovanni’. Most apple cultivars (42.42%) have a Young’s Modulus ranging from 1.0 to 1.5 MPa. In apple fruit, mechanical properties depend largely on its structure and composition and are correlated to different cultivars, and any variations in structure will influence its mechanical properties and hence the texture also during cold storage [53,54].

Furthermore, the browning index in all analyzed apple cultivars showed high variability ranging from 5.63% to 38.30% (Figure 2b). ‘Cusanara’ displayed the highest browning rate, followed by ‘Cannamela’, ‘San Nicola’, ‘Paradiso’, and ‘Zampa di Cavallo’ with values greater than 30%. ‘Martina’ and Tubiona’, as well as ‘Golden B’, displayed a browning index lower than 10% (Figure 2b). Flesh browning is an important qualitative trait that reduces the commercial value of apple fruit after cutting [55]. Due to the high concentration of polyphenols and polyphenol oxidase (PPO) activity, the flesh of apple cultivars quickly becomes dark following cell rupture with differences caused by genetic factors [55,56].

### 2.2. Correlations among the Morphological and Qualitative Traits

Pearson correlation coefficients with positive and negative significant correlations among the morphological and qualitative traits were reported in Appendix A. Tree growth vigor showed positive and significant correlation with fruit diameter (r = 0.444; *p* ≤ 0.01), size of lenticels (r = 0.476; *p* ≤ 0.01) and area of russet around eye basin (r = 0.412; *p* ≤ 0.05), while observed negative correlations with leaf ratio length/width (r = −0.449; *p* ≤ 0.01). Tree habit showed a positive correlation with leaves attitude in relation to the shoot (r = 358; *p* ≤ 0.05) and several fruit features such as weight (r = 0.633; *p* ≤ 0.01), size (r = 0.535; *p* ≤ 0.01), height (r = 0.590; *p* ≤ 0.01), diameter (r = 0.537; *p* ≤ 0.01), number of lenticels (r = 0.492; *p* ≤ 0.01), size of lenticels (r = 0.614; *p* ≤ 0.01), length of the stalk (r = 0.454; *p* ≤ 0.01), and depth of the stalk cavity (r = 0.366; *p* ≤ 0.05), while the type of bearing of 1-year-old shoots was positively correlated with the arrangement of petals in the flower (r = 0.461; *p* ≤ 0.01) and fruit ground color (r = −0.394; *p* ≤ 0.05). The thickness of 1-year-old shoots was positively correlated with length (r = 0.381; *p* ≤ 0.05) and width of leaves (r = 0.414; *p* ≤ 0.05), length of the petiole (r = 0.394; *p* ≤ 0.01), fruit size (r = 0.351; *p* ≤ 0.05) and weight (r = 0.442; *p* ≤ 0.01), number of lenticels (r = 0.362; *p* ≤ 0.05), depth of the stalk cavity (r = 0.444; *p* ≤ 0.01), and time of harvest (r = 0.367; *p* ≤ 0.05) while eating maturity (r = 0.523; *p* ≤ 0.01) was negatively correlated with the bloom of the skin (r = −0.387; *p* ≤ 0.05), and width of stripes (r = −0.372; *p* ≤ 0.05). The internode length in 1-year-old shoots displayed a positive correlation with the number of lenticels (r = 0.353; *p* ≤ 0.05), diameter with petals pressed into a horizontal position (r = 0.363; *p* ≤ 0.05), fruit ribbing (r = 0.521; *p* ≤ 0.01), length of sepal (r = 0.367; *p* ≤ 0.05), while was negatively correlated with the intensity of green color in the leaves (r = −0.378; *p* ≤ 0.05) and thickness of fruit stalk (r = −0.466; *p* ≤ 0.01).

Leaves length showed a positive correlation with leaves width (r = 0.539; *p* ≤ 0.01), length of the petiole (r = 0.632; *p* ≤ 0.01), fruit size (r = 0.344; *p* ≤ 0.05), fruit ground color (r = 0.443; *p* ≤ 0.01), while the width of the fruit stalk cavity (r = 0.420; *p* ≤ 0.05) negatively correlated with the pattern of overcolor (r = −0.400; *p* ≤ 0.05) and width of stripes in the fruit (r = −0.523; *p* ≤ 0.01). Leaves width was positively correlated with fruit size (r = 0.488; *p* ≤ 0.01), fruit ground color (r = −0.443; *p* ≤ 0.01), area of russet on cheeks (r = 0.538; *p* ≤ 0.01), area of russet around eye basin (r = 0.573; *p* ≤ 0.01) and width of fruit stalk cavity (r = 0.528; *p* ≤ 0.01) while was negatively correlated with the pattern of overcolor (r = −634 *p* ≤ 0.05) and the number of lenticels (r = −0.568; *p* ≤ 0.01).

Fruit size was positively correlate with tree habit (r = 0.535; *p* ≤ 0.01), fruit weight (r = 0.821; *p* ≤ 0.01), shoot thickness (r = 0.351; *p* ≤ 0.05), length and width of leaves (r = 0.344; *p* ≤ 0.05; r = 0.488; *p* ≤ 0.01), petiole length (r = 0.396; *p* ≤ 0.05), fruit height and diameter (r = 0.684; *p* ≤ 0.01; r = 0.623; *p* ≤ 0.01, respectively), size of eye (r = 0.648; *p* ≤ 0.01), area of russet around stalk attachment (r = 0.363; *p* ≤ 0.05), area of russet on cheeks (r = 0.490; *p* ≤ 0.01), area of russet around eye basin (r = 0.469; *p* ≤ 0.01), number of lenticels (r = 0.679; *p* ≤ 0.01), size of lenticels (r = 0.398; *p* ≤ 0.05) and color of fruit flesh (r = 0.644; *p* ≤ 0.01). Fruit ground color was negatively correlated with time for harvest and eating maturity (r = −0.433; *p* ≤ 0.05; r = −0.501; *p* ≤ 0.01, respectively),

SSC was positively correlated with the time of the beginning of flowering, for harvest and of eating maturity (r = −0.406; *p* ≤ 0.05; r = −0.532; *p* ≤ 0.01, r = −0.480; *p* ≤ 0.01 respectively), titratable acidity (r = 0.359; *p* ≤ 0.05). TA showed a negative correlation with pH values (r = −0.507; *p* ≤ 0.01). Firmness showed a negative correlation with the hue of overcolor (r = −0.386; *p* ≤ 0.01) and browning index (r = −0.410; *p* ≤ 0.05). Young’s modulus displayed a positive correlation with time of harvest and eating maturing (r = −0.529; *p* ≤ 0.01; r = −0.522; *p* ≤ 0.01, respectively).

In previous studies, the Pearson correlation has been used to evaluate the relationships among bio-agronomic and pomological traits in the germplasm of apples [57], plum and wild plum [58,59], medlar [60] and feijoa [61].

### 2.3. Principal Component Analysis

The principal component analysis (PCA) was carried out using a matrix containing all analyzed parameters to highlight the main differentiating parameters of the variation to make data interpretable and easily visualized. All of the variables in the 2D bi-plot are represented by a vector with different directions and lengths that indicate the contribution of each variable to the 2 principal components in the biplot (Figure 3). The first three components explained a total of 37.36% of the variance, with PC1 accounting for 17.05% of the variance, PC2 accounting for 11.53%, and PC3 accounting for 8.79%. Several tree traits such as tree vigor, habit, type of bearing, the thickness of 1-year-old shoots, internode length, and lenticel number were positively correlated with PC1 as well as some leaves traits such as attitude in relation to shoot length, width, ratio length/width, the intensity of green color, incisions of margin, and the extension of anthocyanin coloration from the base of the petiole. All morphological fruit traits, as well as fruit size, height, and diameter, were positively correlated to PC1, while skin colorimetric traits were related to PC2.

PCA plot displayed that the apple cultivars with proximity were more similar in terms of bio-agronomic and qualitative traits correlated to PC1 and PC2 and were placed in the same quadrant (Figure 3). ‘Ananassa’, ‘Zampa di Cavallo’, San Francesco’, ‘Tubiona’, and ‘Trumuntana’ were positively correlated to PC1 together with 2 standard cultivars, while ‘Austina’, ‘Austegna’, ‘Carne’, ‘Paradiso’, ‘Prete’, ‘Re’, and ‘Zitella’ were negatively correlated to this PC. ‘Acquata’, ‘Aitaniello’, ‘Arancio’, ‘Fragola’, ‘Parrocchiana’, ‘San Giovanni’, ‘Suricillo’, and ‘Latte’ were negatively correlated with PC2 while other apple cultivars were correlated positively at this PC.

Although in breeding programs, only bio-agronomic traits were considered, the depth characterization of apple germplasm considering all UPOV descriptors can be important. Furthermore, agronomic traits are important factors used during markers-assisted selection that help fruit tree breeders in the early identification of interesting apple genotypes [62].

### 2.4. Cluster Analyses

The dendrogram constructed using all morphological and physicochemical traits and based on the Euclidean distance highlighted 2 major clusters, cluster I and II, that define apple groups with similar characteristics (Figure 4). Seven accessions, including ‘Annurca Rossa del Sud’, were grouped in cluster I, while the 26 apple cultivars were placed in cluster II. Each cluster was divided into 2 subclusters; I-A contained 2 cultivars, ‘Tubiona’ and ‘Golden B’, and I-B was further divided into 2 clades that included, in the first, ‘Ananassa’ and ‘Zampa di Cavallo’ and, in the second, ‘Arito’, ‘Martina’ and ‘Trumuntana’. Subcluster II-A contained a single branch containing ‘Annurca Rossa del Sud’ and the other branch grouped eight apple cultivars (‘Agostinella Rossa’, ‘Chianella’, ‘Fragola’, Lazzarola, ‘Paradiso’, ‘San Francesco’, ‘San Nicola’, and ‘Tenerella’). Subcluster II-B contained a single branch including ‘Zitella’, and the other branch grouped 10 apple cultivars (Ambrosio, ‘Cusanara’, ‘Prete’, ‘Arancio’, ‘Latte’, ‘Austegna’, ‘Parrocchiana’, ‘Suricillo’, ‘Vivo’, and ‘Re’), while other clade contained a single branch including ‘San Giovanni’ and the other branch grouped five apple cultivars (‘Carne’, ‘Cannamela’, ‘Austina’, ‘Aitaniello’, and ‘Acquata’).

The relationships of similarity among apple cultivars in the dendrogram (Figure 4) showed a different distribution in the PCA plot (Figure 3). The apple cultivars included in subcluster II-B were positioned along negative PC1 and distributed along PC2, differently from the cultivars in cluster II-A that showed positive PC1. The PCA results confirmed a strong similarity among different apple cultivars, as highlighted by the results of the cluster analysis.

In this apple germplasm collection, a high level of morphological and pomological variabilities was observed. Previous studies have reported high phenotypic diversity of different apple cultivars [44,51,63].

## 3. Materials and Methods

### 3.1. Plant Materials

Thirty-one autochthonous apple cultivars selected from different provinces of the Campania region (Southern Italy) are enrolled in the Campania region germplasm bank and cultivated in the experimental farm ‘‘Areanova’’ in Pignataro Maggiore, Caserta, Italy (41°04′ N. 14°19′ E with an altitude of 61 m above sea). Apple collection was realized in 2013 using 3 trees for each cultivar grafted onto dwarfed M26 rootstock. The plants were trained to palmette training systems and spaced 5 m between the rows and 2.5 m within the row, accounting for a planting density of 800 trees/ha. These local cultivars were analyzed and compared with the ‘Annurca Rossa del Sud’ and ‘Golden B’ cultivars grown in the same field. All apple trees received the same agronomic practices, such as winter pruning, soil amendment in early winter, and the standard preventive application of fungicides and pesticides during different phenological stages. The trees were irrigated every 2 weeks from June to September to reduce water stress during summer. The fruit was harvested at stage 89 of the Biologische Bundesanstalt, Bundessortenamt and CHemical industry (BBCH) scale from each tree per cultivar (3 biological replicates, n = 30) and 3 technical replicates were performed (n = 10).

### 3.2. Morphological Traits

Morphometric analyses were performed using standard apple descriptors established by the Union for the Protection of New Varieties of Plants (UPOV) guidelines (TG/14/9-UPOV 2005) [21]. A set of 57 morphological descriptors were evaluated on 5 organs: 4 descriptors on plants, 5 on 1-year-old winter woody branches (or shoots), 9 on leaves, 5 on flowers, and 34 on fruits. Twenty leaves, 10 flowers, 10 shoots and 50 fruits were randomly hand-collected from specific canopy positions so that the environmental impact was comparable, placed in boxes, transported to the laboratory and for further analysis.

### 3.3. Pomological and Physicochemical Analyses

Thirty fruits from each cultivar were selected for pomological and physical-chemical analysis. The weight of the fruit was determined using a precise digital balance (Practum 213-1S, Sartorius, Göttingen, Germany). Total soluble solids content (SSC) was measured using a digital refractometer (Sinergica Soluzioni, DBR35, Pescara, Italy), and the results were expressed as °Brix. Acid-base titration of apple juice was performed using a digital pH meter (Model 2001, Crison, Barcelona, Spain) to measure the titratable acidity (TA) through NaOH 0.1 N until the end point of pH 8.1. The results were expressed as grams of malic acid equivalent per liter of juice (g malic acid L^−1^). The pH value of juice from each cultivar was also measured using the same digital pH meter at 20 °C. Fruit firmness (expressed as kg/cm^2^) was measured after removing the skin with a digital penetrometer equipped with an 11 mm tip (TR snc, Forlì, Italy). The skin color of fruits was assessed using a Minolta colorimeter (CR5. Minolta Camera Co. Japan) to determine chromaticity values L* (Lightness), a* (green to red), and b* (blue to yellow).

Rheologic analyses were performed by Dynamometer Mod. LRX plus (Ametek, Inc., Berwyn, PA, USA; Lloyd Instruments, Bognor Regis, UK). Ten samples per cultivar were put through a puncture test using a Volodkevitch model FG/VBS device and a load of 100 N. In accordance with the American Society for Testing and Materials (ASTM), Young’s modulus was calculated [64]. Enzymatic browning was determined by measuring the color of apples’ inner surface using a colorimeter (CR-200 Minolta, Japan) after 240 min of exposure to air and room temperature. The enzymatic browning was calculated by converting the Hunter scale (Lab) values into whiteness index (WI) through the formula:(1)$$\mathrm{WI}=100-\sqrt{{\left(100-L\right)}^{2}}+{\;a}^{2}+{\;b}^{2}$$

The results were expressed as the percentage of browning (%).

### 3.4. Statistical Analysis

All data are presented as mean ± standard deviation (SD). Statistical significance among the apple cultivars was evaluated using a 1-way analysis of variance (ANOVA) through Duncan’s test using the SPSS software (version 20.0; SPSS Inc., Chicago, IL, USA). Omega squared (ω^2^) was used as the measure of effect size. The different letters represent differences that were deemed significant at *p* < 0.05. The coefficient of variation (CV) was calculated to evaluate the variability for each determined trait. Pearson’s correlation coefficient (*p* < 0.05, *p* < 0.01 was calculated to display the correlation among analyzed traits) with SPSS^®^ software version 20 (SPSS Inc., Chicago, IL, USA, Norusis, 1998). The cluster analysis, based on Euclidean distance, and principal component analysis to evaluate relationships among apple cultivars were performed using OriginPro 2015 software.

## 4. Conclusions

Currently, there is increasing interest in the preservation of the autochthonous genetic heritage of fruit species. The ex-situ apple germplasm collection located in the Campania region, characterized in this study, represents an irreplaceable genetic resource with considerable morphological and pomological variabilities among several cultivars. Our findings demonstrated a high degree of phenotypic diversity for analyzed traits.

Nowadays, some local cultivars, widespread only in restricted geographical areas, could be reintroduced in cultivation contribution to improving the diversity of our diets and contemporary to preserve knowledge on traditional agricultural systems.

However, the conservation of autochthonous cultivars is not enough; these resources should be accessible to people, breeders, or farmers through participatory action research and, in this way, can act as a catalyst for transformative food systems.

## Figures and Tables

**Figure 1 plants-12-01160-f001:**
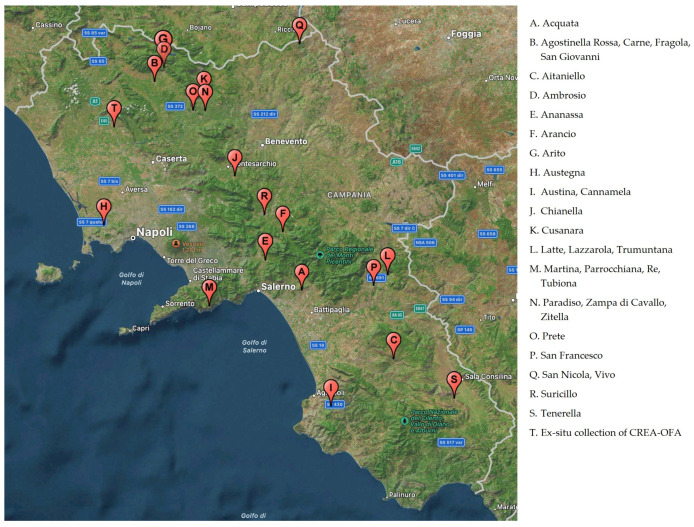
Geographical location of the original place of several cultivars enrolled in the inventory of Campania regional bank and the ex-situ collection of CREA-OFA (Pignataro Maggiore, Caserta Italy).

**Figure 2 plants-12-01160-f002:**
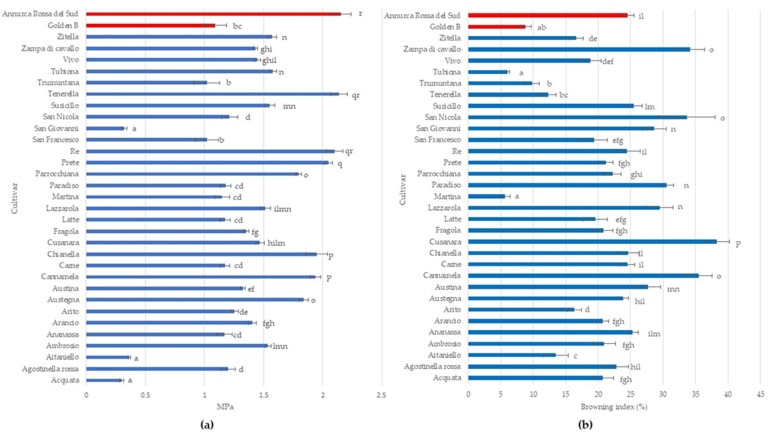
Young’s module (**a**) and enzymatic browning (**b**) in the 31 autochthonous apple cultivars (Acquata, Agostinella rossa, Aitaniello, Ambrosio, Ananassa, Arancio, Arito, Austegna, Austina, Cannamela, Carne, Chianella, Cusanara, Fragola, Latte, Lazzarola, Martina, Paradiso, Parrocchiana, Prete, Re, San Francesco, San Giovanni, San Nicola, Suricillo, Tenerella, Trumuntana, Tubiona, Vivo, Zampa di Cavallo, and Zitella) compared to two commercial cultivars (Golden B and Annurca Rossa del Sud). Data represent means ± SD. The same letters indicate non-significant differences (Duncan test) between cultivars (*p* < 0.05).

**Figure 3 plants-12-01160-f003:**
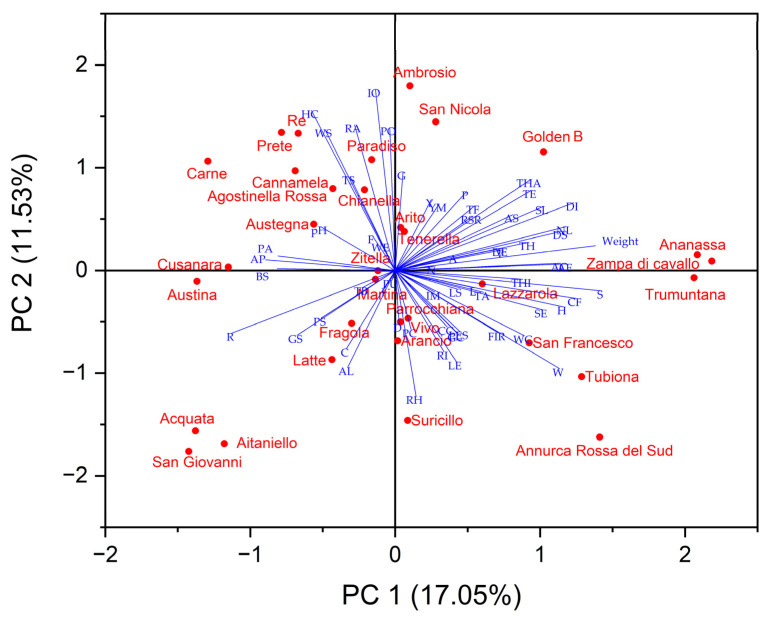
2-D Biplot of morphological, pomological and physicochemical parameters in the thirty-one autochthonous apple cultivars and two commercial cultivars (‘Golden B’ and ‘Annurca Rossa del Sud’). (Titratable acidity: TA; Firmness: FIR; Browning index: BI; Young’s module: YM; Solid soluble content: SSC; Type: T; Only varieties with ramified tree type-habit: TH; Type of bearing: TB; Thickness: THI; Length of internode: L; Color on sunny side: C; Pubescence (on distal half of shoot): P; Number of lenticels: N; Attitude in relation to shoot: A; Length: LE; Width: W; Ratio length/width: R; Intensity of green color: I; Incisions of margin (upper half): IM; Pubescence on lower side: PU; Petiole—length: PL; Petiole—extent of anthocyanin coloration from base: PA; Predominant color at balloon stage: PC; Diameter with petals pressed into horizontal position: D; Arrangement of petals: AP; Position of stigmas relative to anthers: PS; Young fruit- extent of anthocyanin overcolor: Y; Size: S; Height: H; Diameter: DI; Ratio height/diameter: RH; General shape: G; Ribbing: RI; Crowning at calyx end: CC; Size of eye: SE; Length of sepal: LS; Bloom of skin: BS; Greasiness of skin: GS; Ground color: GC; Relative area of overcolor: RA; Hue of overcolor—with bloom removed: HC; Intensity of overcolor: IO; Pattern of overcolor: PO; Width of stripes: WS; Area of russet around stalk attachment: AS; Area of russet on cheeks: AC; Area of russet around eye basin: AE; Number of lenticels: NL; Size of lenticels: SL; Length of stalk: LES; Thickness of stalk: TS; Depth of stalk cavity: DS; Width of stalk cavity: WC; Depth of eye basin: DE; Width of eye basin: WE; Firmness of flesh: F; Color of flesh: CF; Aperture of locules (in transverse section): AL; Time of beginning of flowering: TF; Time for harvest: THA; Time of eating maturity: TE).

**Figure 4 plants-12-01160-f004:**
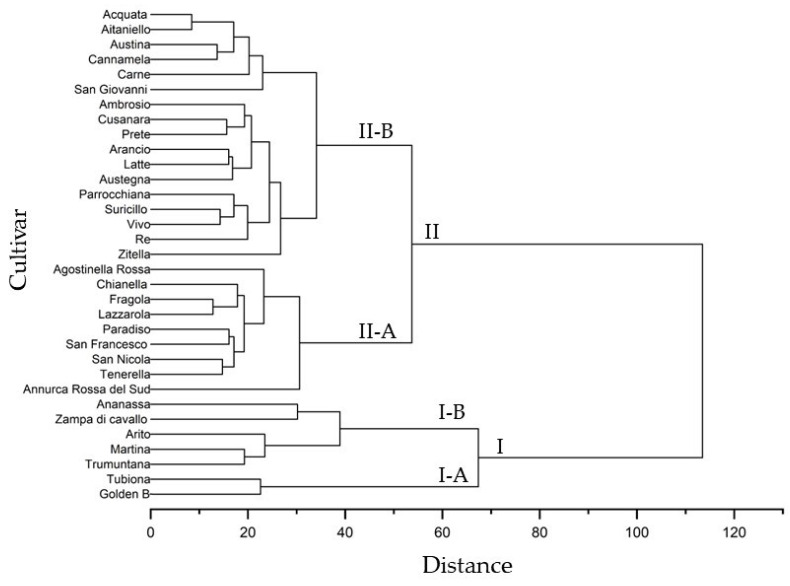
Cluster analysis of morphological, pomological and physicochemical parameters in the thirty-one autochthonous apple cultivars and two commercial cultivars (‘Golden b’ and ‘Annurca Rossa del Sud’).

**Table 1 plants-12-01160-t001:** Descriptive statistics for morphological traits utilized in autochthonous apple cultivars compared to standard cultivars (‘Annurca Rossa del Sud’ and ‘Golden B’).

No.	Organ	Character	Abbreviation	Unit	Min.	Max.	Mean	SD	CV%
1	Tree	Vigor	V	Code	1	7	5.48	1.23	22.39
2		Type	T	Code	2	2	2	0	0
3		Only varieties with ramified tree type: habit	TH	Code	1	3	1.51	0.62	40.82
4		Type of bearing	TB	Code	1	3	2.24	0.75	33.50
5	One-year-old shoot	Thickness	THI	Code	3	9	4.33	1.47	33.97
6		Length of internode	L	Code	1	7	4.57	1.20	26.22
7		Color on the sunny side	C	Code	2	4	2.39	0.70	29.43
8		Pubescence (on the distal half of the shoot)	P	Code	1	7	4.51	1.66	36.78
9		Number of lenticels	N	Code	3	7	3.90	1.23	31.57
10	Leaf	Attitude in relation to shoot	A	Code	1	3	1.12	0.41	37.03
11		Length	LE	Code	1	7	4.06	1.97	48.45
12		Width	W	Code	3	7	4.51	1.50	33.28
13		Ratio length/width	R	Code	3	7	5.60	1.46	25.98
14		Intensity of the green color	I	Code	3	7	4.93	0.61	12.33
15		Incisions of margin (upper half)	IM	Code	1	5	3.12	0.99	31.80
16		Pubescence on the lower side	PU	Code	1	3	1.57	0.56	35.58
17		Petiole: length	PL	Code	3	7	5.09	1.70	33.44
18		Petiole: the extent of anthocyanin coloration from base	PA	Code	3	7	3.79	1.32	34.77
19	Flower	Predominant color at the balloon stage	PC	Code	1	5	3.27	0.98	29.85
20		Diameter with petals pressed into a horizontal position	D	Code	5	7	5.06	0.35	6.88
21		Arrangement of petals	AP	Code	1	3	1.67	0.69	41.53
22		Position of stigmas relative to anthers	PS	Code	1	3	2.39	0.70	29.43
23	Fruit	Young fruit: the extent of anthocyanin overcolor	Y	Code	1	9	1.42	1.48	103.89
24		Size	S	Code	1	9	3.58	2.95	82.44
25		Height	H	Code	3	7	4.21	1.32	31.27
26		Diameter	DI	Code	3	7	4.70	1.42	30.33
27		Ratio height/diameter	RH	Code	1	9	4.15	2.87	69.22
28		General shape	G	Code	2	7	5.06	2.19	43.33
29		Ribbing	RI	Code	1	2	1.06	0.24	22.85
30		Crowning at the calyx end	CC	Code	1	2	1.24	0.44	35.03
31		Size of eye	SE	Code	3	7	4.88	1.58	32.31
32		Length of sepal	LS	Code	3	7	3.36	0.93	27.63
33		Bloom of skin	BS	Code	1	3	1.36	0.55	40.24
34		Greasiness of skin	GS	Code	1	2	1.30	0.47	35.82
35		Ground color	GC	Code	2	6	4.76	0.66	13.93
36		Relative area of overcolor	RA	Code	1	9	3.06	2.57	84.07
37		Hue of overcolor—with bloom removed	HC	Code	2	3	2.60	0.50	19.33
38		Intensity of overcolor	IO	Code	3	5	4.27	0.98	23.05
39		Pattern of overcolor	PO	Code	1	7	2.36	1.84	77.84
40		Width of stripes	WS	Code	3	7	4.82	1.40	29.08
41		Area of russet around stalk attachment	AS	Code	1	3	1.45	0.71	48.89
42		Area of russet on cheeks	AC	Code	1	3	1.24	0.61	49.41
43		Area of russet around eye basin	AE	Code	1	3	1.30	0.68	52.49
44		Number of lenticels	NL	Code	3	7	4.58	1.56	34.13
45		Size of lenticels	SL	Code	3	7	4.45	1.75	39.32
46		Length of stalk	LES	Code	1	7	3.67	1.63	44.54
47		Thickness of stalk	TS	Code	3	7	5.00	1.11	22.36
48		Depth of stalk cavity	DS	Code	3	7	4.81	1.53	31.75
49		Width of stalk cavity	WC	Code	3	7	4.21	1.49	35.49
50		Depth of eye basin	DE	Code	3	7	4.39	1.17	26.65
51		Width of eye basin	WE	Code	3	7	4.33	1.38	31.95
52		Firmness of flesh	F	Code	3	9	6.09	1.67	27.34
53		Color of flesh	CF	Code	1	2	1.15	0.36	31.62
54		Aperture of locules (in transverse section)	AL	Code	1	3	1.54	0.71	46.01
55	Flower	Time of beginning of flowering	TF	Code	3	9	5.67	1.47	25.98
56	Fruit	Time for harvest	THA	Code	1	9	5.84	2.24	38.26
57		Time of eating maturity	TE	Code	1	9	6.12	2.30	37.60

**Table 2 plants-12-01160-t002:** Bio-agronomic traits and qualitative traits such as firmness, solid soluble content (SSC), titratable acidity (TA), and pH of juice in autochthonous apple cultivars compared to standard cultivar (‘Annurca Rossa del Sud’ and ‘Golden B’).

Cultivar	Growing Area (Italy)	Harvest Time	Fruit Weight (g)	Ground Colour	Overcolour	Firmness (Kg/cm^2^)	SSC (°Brix)	TA (g Malic Acid L^−1^)	pH
Acquata	Montecorvino Pugliano (SA)	Late June	42.40 ± 5.69 (abc)	Green-Yellow	-	7.64 ± 0.53 (ghilm)	9.27 ± 0.70 (cd)	2.36 ± 0.40 (a)	3.61 ± 0.05 (ilm)
Agostinella Rossa	Alife (CE)	Late August	107.80 ± 13.06 (i)	Whitish-Green	Red	7.32 ± 1.96 (fghil)	12.60 ± 0.50 (lmn)	4.30 ± 0.60 (de)	2.90 ± 0.05 (abc)
Aitaniello	Ottati (SA)	Late June	37.90 ± 11.12 (ab)	Green-Yellow	-	7.25 ± 0.74 (fghil)	9.00 ± 0.40 (bc)	3.01 ± 0.57 (bc)	3.58 ± 0.09 (ilm)
Ambrosio	Castello Matese (CE)	Late October	76.60 ± 8.96 (g)	Whitish-Green	Pink-red	9.28 ± 1.43 (op)	12.00 ± 0.40 (hil)	3.00 ± 0.81 (bc)	3.52 ± 0.07 (hilm)
Ananassa	Fisciano (SA)	Late September	178.70 ± 43.85 (n)	Green-Yellow	-	8.76 ± 1.14 (no)	8.20 ± 0.50 (a)	4.03 ± 0.32 (d)	3.73 ± 0.05 (mn)
Arancio	Serino (AV)	Early October	110.00 ± 14.48 (fg)	Green-Yellow	-	7.96 ± 1.92 (hilmn)	13.80 ± 0.70 (o)	4.69 ± 0.24 (ef)	3.69 ± 0.18 (lmn)
Arito	San Gregorio Matese (CE)	Early October	149.17 ± 16.13 (l)	Whitish-Green	Red	8.59 ± 1.09 (no)	13.30 ± 0.50 (no)	3.35 ± 0.46 (c)	3.39 ± 0.19 (fghi)
Austegna	Quarto (NA)	Late August	61.00 ± 8.86 (ef)	Green-Yellow	Red	7.29 ± 1.73 (fghil)	11.56 ± 0.40 (gh)	4.69 ± 0.16 (ef)	3.10 ± 0.09 (cde)
Austina	Torchiara (SA)	Mid August	46.93 ± 10.40 (bcd)	Green-Yellow	Pink-red	6.55 ± 1.38 (def)	8.40 ± 0.30 (ab)	3.01 ± 0.53 (bc)	3.67 ± 0.10 (lmn)
Cannamela	Torchiara (SA)	Mid September	46.60 ± 7.77 (bcd)	Whitish-Green	Pink-red	5.39 ± 0.83 (abc)	10.50 ± 0.70 (ef)	4.13 ± 0.38 (de)	4.21 ± 0.25 (pq)
Carne	Alife (CE)	Mid August	31.33 ± 5.71 (a)	Green-Yellow	Red	5.61 ± 2.05 (bc)	11.90 ± 0.42 (hil)	2.34 ± 0.19 (a)	4.05 ± 0.03 (opq)
Chianella	San Martino Valle Caudina (BN)	Mid September	105.80 ± 10.61 (hi)	Green-Yellow	Red	7.05 ± 1.09 (fgh)	14.64 ± 0.30 (p)	7.37 ± 0.29 (i)	3.04 ± 0.08 (fghil)
Cusanara	San Lorenzello (BN)	Early September	77.50 ± 17.04 (g)	Whitish-Green	-	4.62 ± 0.68 (a)	10.60 ± 0.5 (ef)	2.68 ± 0.11 (ab)	3.69 ± 0.09 (hilmn)
Fragola	Alife (CE)	Mid September	94.28 ± 19.06 (g)	Green-Yellow	Pink-red	6.83 ± 1.57 (efg)	11.60 ± 0.30 (ghi)	5.02 ± 0.22 (fg)	3.23 ± 0.08 (defg)
Latte	Colliano (SA)	Early October	63.50 ± 7.24 (ef)	Whitish-Green	-	7.67 ± 1.29 (ghilm)	12.00 ± 0.50 (hil)	8.04 ± 0.32 (l)	3.45 ± 0.21 (ghilm)
Lazzarola	Colliano (SA)	Late September	97.80 ± 12.38 (h)	Green	Pink-red	7.32 ± 0.88 (fghil)	11.40 ± 0.20 (gh)	5.36 ± 0.22 (g)	3.28 ± 0.19 (defgh)
Martina	Agerola (NA)	Late August	156.00 ± 18.77 (m)	Green-Yellow	Pink-red	9.62 ± 1.21 (pq)	13.00 ± 0.40 (mno)	5.10 ± 0.30 (fg)	3.73 ± 0.05 (mn)
Paradiso	Castelvenere (BN)	Late September	103.10 ± 16.16 (hi)	Green-Yellow	Red	8.17 ± 1.10 (lmn)	8.60 ± 0.40 (abc)	2.34 ± 0.49 (a)	5.28 ± 0.15 (r)
Parrocchiana	Agerola (NA)	Late September	52.40 ± 5.69 (cde)	Green-Yellow	-	8.35 ± 1.28 (mn)	11.90 ± 0.20 (hil)	2.34 ± 0.12 (a)	3.04 ± 0.15 (bcd)
Prete	San Salvatore Telesino (BN)	Late September	70.27 ± 11.30 (fg)	Green-Yellow	Red	4.96 ± 0.85 (ab)	12.50 ± 0.25 (lm)	2.36 ± 0.34 (a)	4.27 ± 0.04 (q)
Re	Agerola (NA)	Late October	56.70 ± 8.99 (def)	Whitish-Green	Red	6.94 ± 0.59 (fg)	11.50 ± 0.70 (gh)	7.60 ± 0.20 (il)	2.90 ± 0.06 (abc)
San Francesco	Oliveto Citra (SA)	Early October	109.40 ± 10.09 (i)	Green-Yellow	-	9.95 ± 1.03 (pq)	10.20 ± 0.35 (e)	5.02 ± 0.23 (fg)	2.67 ± 0.15 (a)
San Giovanni	Alife (CE)	Late June	32.90 ± 4.14 (a)	Green-Yellow	-	8.07 ± 0.71 (lmn)	8.00 ± 0.30 (a)	3.35 ± 0.14 (c)	3.57 ± 0.31 (ilm)
San Nicola	Castelvetere Valfortore (BN)	Early October	107.60 ± 9.03 (i)	Green-Yellow	Red	5.90 ± 0.53 (cd)	13.50 ± 0.50 (o)	4.67 ± 0.21 (ef)	4.01 ± 0.20 (op)
Suricillo	Avellino	Mid September	62.20 ± 8.87 (ef)	Green-Yellow	-	10.24 ± 1.39 (q)	11.00 ± 0.40 (fg)	2.68 ± 0.82 (ab)	4.30 ± 0.30 (q)
Tenerella	Sassano (SA)	Late August	109.40 ± 19.94 (i)	Green-Yellow	Red	7.13 ± 1.12 (fghi)	9.90 ± 0.25 (de)	5.00 ± 0.34 (efg)	3.33 ± 0.31 (efghi)
Trumuntana	Colliano (SA)	Early August	156.30 ± 35.20 (m)	Green-Yellow	-	9.28 ± 0.84 (nopq)	12.50 ± 0.20 (lm)	6.70 ± 0.30 (h)	2.85 ± 0.09 (abc)
Tubiona	Agerola (NA)	Mid August	236.00 ± 31.63 (q)	Green-Yellow	-	6.07 ± 1.21 (cde)	12.40 ± 0.10 (ilm)	4.36 ± 0.73 (de)	3.18 ± 0.10 (def)
Vivo	Castelvetere Valfortore (BN)	Late August	59.60 ± 5.53 (def)	Green-Yellow	Pink-red	7.31 ± 1.28 (fghil)	10.60 ± 0.30 (ef)	10.38 ± 0.63 (n)	2.86 ± 0.03 (abc)
Zampa di cavallo	Castelvenere (BN)	Mid September	194.70 ± 43.61 (o)	Green-Yellow	-	8.40 ± 1.21 (mn)	13.00 ± 0.30 (mno)	5.10 ± 0.30 (fg)	2.80 ± 0.10 (ab)
Zitella	Castelvenere (BN)	Late September	79.20 ± 3.83 (g)	Green-Yellow	Red	9.39 ± 1.12 (opq)	10.20 ± 0.30 (e)	3.02 ± 0.25 (bc)	4.20 ± 0.10 (pq)
Annurca Rossa del Sud	-	Late September	122.62 ± 11.97 (p)	Green-Yellow	Red	8.03 ± 1.48 (ilmn)	13.50 ± 0.36 (hil)	9.00 ± 0.65 (m)	3.47 ± 0.02 (ghilm)
Golden B	-	Mid September	223.75 ± 40.94 (l)	Green-Yellow	-	6.86 ± 0.19 (efg)	12.00 ± 0.70 (o)	4.40 ± 0.40 (de)	3.89 ± 0.04 (no)

Values in a column with a common letter are not significantly different, *p* < 0.05 (Duncan Test).

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
