# Peer review of "Autochthonous Apple Cultivars from the Campania Region (Southern Italy): Bio-Agronomic and Qualitative Traits"

_plants, 2023, doi:10.3390/plants12051160_

Round 1
Reviewer 1 Report
Excellent work and impressive apple genetic resource preserved. Congratulations!
Few suggestions: it would be interesting for further research on this germplasm to add few ECPGR traits, considering for instance skin thickness of the fruits, storability traits and tolerance to different key pests and diseases. These ones I consider will be very useful too for breeders.
In the materials and methods section, you can add more information related to the work done in the field or lab. I mentioned only the skin color determination or flesh thickness.

Author Response
Excellent work and impressive apple genetic resource preserved. Congratulations!
Few suggestions: it would be interesting for further research on this germplasm to add few ECPGR traits, considering for instance skin thickness of the fruits, storability traits and tolerance to different key pests and diseases. These ones I consider will be very useful too for breeders.
In a subsequent paper we will certainly evaluate these traits as well, we thank the reviewer for the valuable suggestions.
In the materials and methods section, you can add more information related to the work done in the field or lab. I mentioned only the skin color determination or flesh thickness.
This information has been added in the revised manuscript.
Line 204. Are there any data collected for the apple skin thickness? Is there any correlation with the firmness and storability revealed?
Thank you for this consideration. In a new postharvest study has been tested the storability of these apple cultivars and we have measured skin thickness to correlate this trait to firmness and storage time.
Line 368. The plants in the row.
Done
Line 384. Please add the firmness device used.
This information has been added in the revised manuscript.
Line 384.How was discriminated apples by skin colour. What device was used?
This information has been added in the revised manuscript.

Reviewer 2 Report
The object of the work was to analyze thirty-one local apple cultivars of Campania 17 region (Southern Italy) using agronomic, morphological (UPOV descriptors) and physico-chemical traits. However, the scientific progress is not clearly stated. The statement that ‘This apple germplasm collection represents an irreplaceable genetic resource with considerable morphological and pomological variabilities among several cultivars’ as well as ‘some local cultivars, widespread only in restricted geographical areas, could be reintroduced in cultivation contribution to improve the diversity of our diets and contemporary to preserve knowledge on traditional agricultural systems’. The findings demonstrate a high degree of phenotypic diversity for analyzd traits, but only referred to 2 commercial cultivars, and there are no evidences of the diversity without no supplementary analysis such as genotypic analysis.
The main critical point is that the authors do not describe exactly the time point of harvest. At fruit maturity, the firmness and acid content steadily decrease, meanwhile, the concentration of sugars and soluble carbohydrates is increasing. The dynamics need to establish a homologous stage and some sampling criteria for an impartial comparison of genotypes referred to the physicochemical analyzes of the fruit. Without common baseline criteria, results may not reflect genotype-dependent variation.
The collection of local apple cultivars is a valuable gen pool which can be exploited in propagation for local use or breeding. How can these compiled data be used for progressing research? A statement is needed
Supplementary comments:
Line 59. What does it mean?: ‘They (local germplasm) exhibit good local environmental condition and are genetic resources with high genetic variability in crops’
Line 108. Results on vigor. If ‘the plants were trained to palmette training systems’, the vigor is controlled by the pruning system. It should be explained.
Line 210. ‘The results showed that some traditional cultivars have resistance that is statistically similar, if not higher, than standard cultivars’. Only referred to 2 commercial cultivars. ‘This means that most apple traditional cultivars are resistant to mechanical stresses or normal physiological processes as reported by Panzella et al.' Panzella’s work is based on the study of nutraceutical properties of little widespread local apple cultivars, in terms of phenolic composition and free radical scavenging activity, not in mechanical stresses.
Line 214: ‘Firmness variations is due only genetic variability among analyzed cultivar being cultivated in the same pedoclimatic conditions’. And the samples must be taken with a definable time point of harvest. Firmness may reflect different stages of maturity.
Line 232:Is por in
Line 234: ‘…the flesh of all apple cultivars…’. Not all
Lines 235 – 238: Out of place…. Not relevant.
Line 249: Pearson correlation coefficients should be shown in the document
Line 298: Principal component analysis. PC1 and PC2 must be explained in text and in the caption. The results are not clear. For example: ‘If Dimension 1 (PC1) is xxxxx, the distribution obtained suggests that xxxxx’. Or ‘The grouping of variables in Dimension 2 (PC2) includes xxx’
Line 362: What does it mean?: Thirty-one autochthonous apple cultivars selected from different provinces of Campania region (Southern Italy) are enrolled in the Campania region germplasm bank and cultivated in the experimental farm ''Areanova'' in Pignataro Maggiore, Caserta, Italy on ten-year-old trees cultivar grafted on to dwarfed M26 rootstock. The sentence must be clarified; the autochthonous cultivars were cultivated on ten-year-old trees??
Author Response
The object of the work was to analyze thirty-one local apple cultivars of Campania 17 region (Southern Italy) using agronomic, morphological (UPOV descriptors) and physico-chemical traits. However, the scientific progress is not clearly stated. The statement that ‘This apple germplasm collection represents an irreplaceable genetic resource with considerable morphological and pomological variabilities among several cultivars’ as well as ‘some local cultivars, widespread only in restricted geographical areas, could be reintroduced in cultivation contribution to improve the diversity of our diets and contemporary to preserve knowledge on traditional agricultural systems’. The findings demonstrate a high degree of phenotypic diversity for analyzd traits, but only referred to 2 commercial cultivars, and there are no evidences of the diversity without no supplementary analysis such as genotypic analysis.
Ex-situ germplasm collection represents an important source to preserve the genetic diversity and agronomic characteristics. We have ongoing genetic analyses that will correlate with agronomic traits in the next study.
The main critical point is that the authors do not describe exactly the time point of harvest. At fruit maturity, the firmness and acid content steadily decrease, meanwhile, the concentration of sugars and soluble carbohydrates is increasing. The dynamics need to establish a homologous stage and some sampling criteria for an impartial comparison of genotypes referred to the physicochemical analyzes of the fruit. Without common baseline criteria, results may not reflect genotype-dependent variation.
All apple cultivars were harvested at stage 89 of BBCH scale (Fruit ripe for consumption). Our research group has analysed this apple collection since its entered full production. This information has been addede in the revised manuscript.
The collection of local apple cultivars is a valuable gen pool which can be exploited in propagation for local use or breeding. How can these compiled data be used for progressing research? A statement is needed
Some of these cultivars could be valorised but agricultural policies play a central role, in particular, those of rural development, which can, if properly set up, promote the link between tradition and modernity, avoiding interruptions and using agrobiodiversity as a factor in local development. Furthermore, some of these cultivars could be used in breeding programme for their agronomic traits (time of flowering and harvest) considering the climatic changes of the last years. For this reason, it is not only a simple implementation of conservation policies for plant genetic resources, but also a change of perspective by moving towards a system of safeguarding to provide a reciprocal interaction and a necessary complementary action between ex situ and in situ/on-farm conservation.
Supplementary comments:
Line 59. What does it mean? ‘They (local germplasm) exhibit good local environmental condition and are genetic resources with high genetic variability in crops’
Autochthonous cultivars represent the local germplasm cultivated mainly in the marginal areas and they show a good adaptability to the local environment and represent a valuable source for the crop genetic variability. These sentence has been changed.
Line 108. Results on vigor. If ‘the plants were trained to palmette training systems’, the vigor is controlled by the pruning system. It should be explained.
The apple cultivars were grafted on the same rootstock and grown with the same training system to assess their vigor.
Line 210. ‘The results showed that some traditional cultivars have resistance that is statistically similar, if not higher, than standard cultivars’. Only referred to 2 commercial cultivars. ‘This means that most apple traditional cultivars are resistant to mechanical stresses or normal physiological processes as reported by Panzella et al.' Panzella’s work is based on the study of nutraceutical properties of little widespread local apple cultivars, in terms of phenolic composition and free radical scavenging activity, not in mechanical stresses.
In the paper of “Panzella et al., 2013” have been analyzed bio-agronomic and qualitative traits in local Italian apple cultivars. The authors reported this sentence at the end of section 3.1.
Line 214: ‘Firmness variations is due only genetic variability among analyzed cultivar being cultivated in the same pedoclimatic conditions’. And the samples must be taken with a definable time point of harvest. Firmness may reflect different stages of maturity.
The apples were harvested at the same stage of BBCH scale (89)
Line 232: Is por in
Done
Line 234: ‘…the flesh of all apple cultivars…’. Not all
Done
Lines 235 – 238: Out of place…. Not relevant.
This sentence has been deleted.
Line 249: Pearson correlation coefficients should be shown in the document
Many parameters have been evaluated by Pearson correlations. Consequently, the table with correlations is given in Supplementary Material Table S5 as Excel file. If we insert the table in the manuscript the values are not readable.
Line 298: Principal component analysis. PC1 and PC2 must be explained in text and in the caption. The results are not clear. For example: ‘If Dimension 1 (PC1) is xxxxx, the distribution obtained suggests that xxxxx’. Or ‘The grouping of variables in Dimension 2 (PC2) includes xxx’
This section has been improved in the revised manuscript.
Line 362: What does it mean?: Thirty-one autochthonous apple cultivars selected from different provinces of Campania region (Southern Italy) are enrolled in the Campania region germplasm bank and cultivated in the experimental farm ''Areanova'' in Pignataro Maggiore, Caserta, Italy on ten-year-old trees cultivar grafted on to dwarfed M26 rootstock. The sentence must be clarified; the autochthonous cultivars were cultivated on ten-year-old trees??
This sentence has been modified.

Reviewer 3 Report
Dear Authors,
Thank you for well written and interesting to read manuscript. My main remarks are to the methodology part and the quality of supplement tables.
It is not clear how many trees of each cultivar did you test?
How many replicates were there?
If you sampled 30 fruits for quality analysis, did you replicate them?
The tables should be self explanatory. However, table S2 and S4 is impossible to understand.
Please, explain all abbreviations and numerical notes.
Please, explain all UPOV numbers (do not ask the reader to look after UPOV guidelines).
Rearrange Table S1. Starting from the page 2 it is impossible to follow to what variety one or another descriptor belongs.
Author Response
Dear Authors,
Thank you for well written and interesting to read manuscript. My main remarks are to the methodology part and the quality of supplement tables.
It is not clear how many trees of each cultivar did you test?
This information has been added in the revised manuscript.
How many replicates were there?
This information has been added in the revised manuscript.
If you sampled 30 fruits for quality analysis, did you replicate them?
This information has been added in the revised manuscript.
The tables should be self explanatory. However, table S2 and S4 is impossible to understand.
The number of Table has been changes and table S2 now is Table S3. The table S3 have been modified. The table S5 has been uploaded as Excel file in supplementary materials.
Please, explain all abbreviations and numerical notes.
All abbreviations and numerical notes have been added.
Please, explain all UPOV numbers (do not ask the reader to look after UPOV guidelines).
A table with abbreviations has been added (Table S1)
Rearrange Table S1. Starting from the page 2 it is impossible to follow to what variety one or another descriptor belongs.
The table has been modified, now is Table S2.

Round 2
Reviewer 2 Report
The authors have clarified and modified the issues raised.
Regarding my comments, the observations made on the content of the work have been corrected and are acceptable for publication